# SUDOKU-BENCH: Evaluating creative reasoning with Sudoku variants

**Jeffrey Seely**[1]
jeffrey@sakana.ai

**Yuki Imajuku**[1]
imajuku@sakana.ai

**Tianyu Zhao**[1]
tianyu@sakana.ai

**Edoardo Cetin**[1]
edo@sakana.ai

**Llion Jones**[1]
llion@sakana.ai

[1]Sakana AI, Japan

## Abstract

Existing reasoning benchmarks for large language models (LLMs) frequently fail to capture authentic creativity, often rewarding memorization of previously observed patterns. We address this shortcoming with SUDOKU-BENCH, a curated benchmark of challenging and unconventional Sudoku variants specifically selected to evaluate creative, multi-step logical reasoning. Sudoku variants form an unusually effective domain for reasoning research: each puzzle introduces unique or subtly interacting constraints, making memorization infeasible and requiring solvers to identify novel logical breakthroughs ("break-ins"). Despite their diversity, Sudoku variants maintain a common and compact structure, enabling clear and consistent evaluation. SUDOKU-BENCH includes a carefully chosen puzzle set, a standardized text-based puzzle representation, and flexible tools compatible with thousands of publicly available puzzles—making it easy to extend into a general research environment. Baseline experiments show that state-of-the-art LLMs solve fewer than 15% of puzzles unaided, highlighting significant opportunities to advance long-horizon, strategic reasoning capabilities.

## 1   Introduction

Large-scale language models excel at short-form deduction [12, 29], yet genuinely *creative* reasoning remains elusive. Many standard benchmarks, where current models already rival or surpass human performance [8, 22, 6], often reward the memorization of solution templates [2]. Once these templates are implicitly memorized, incremental accuracy gains offer limited insight into a model's capacity for novel reasoning. Benchmarks such as ARC [3] effectively resist memorization; however, their solutions, while novel to models, remain straightforward for humans, insufficiently capturing the depth of human creative reasoning.

We propose Sudoku variants (Fig. 1) as a unique domain addressing this gap. A Sudoku variant is a logical puzzle defined by a partially filled $n \times n$ grid, accompanied by visual constraints and even a problem-specific set of rules that can only be described in natural language. Yet, each puzzle still admits a unique solution—an $n \times n$ grid fulfilling its constraints. Puzzle creators introduce original rules or combine common constraints in novel ways. Hundreds of user-submitted Sudoku variants are published daily on platforms like Logic Masters Germany [1], deliberately designed to require *creative* insights and subtle logical breakthroughs. Such puzzles precisely target the type of novel, multi-step reasoning that memorization-focused and even popular reasoning benchmarks fail to consistently measure [31].

Submitted to 39th Conference on Neural Information Processing Systems (NeurIPS 2025). Do not distribute.

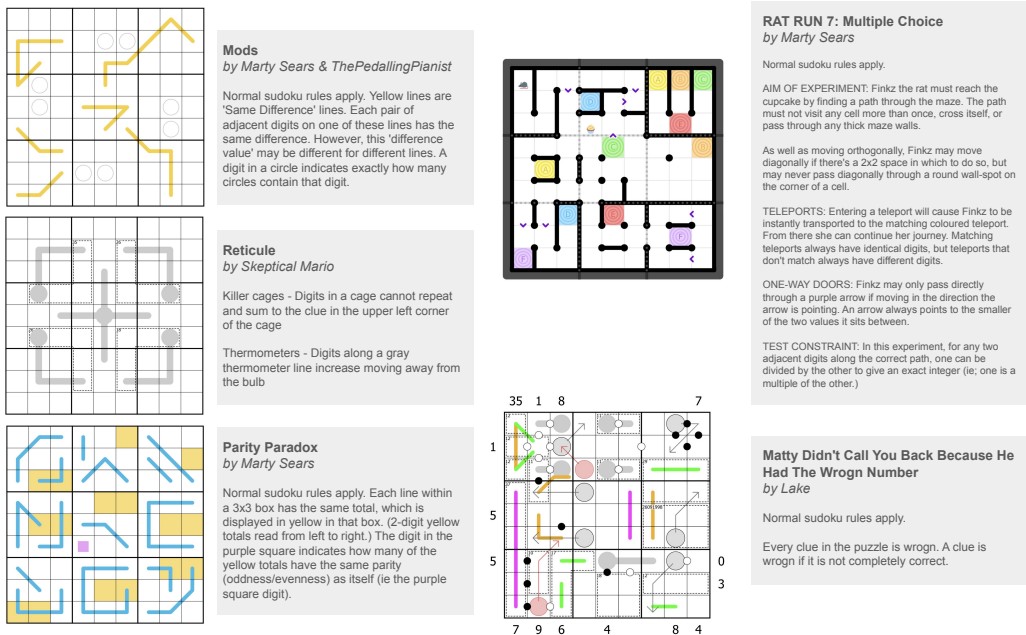

Figure 1: Each Sudoku variant has a unique set of constraints explicitly described in the puzzle rules. Puzzles may feature whimsical rules such as in *Rat Run*, or meta-level constraints, such as requiring all standard Sudoku rules to be intentionally violated.

This paper's contribution is twofold. First, we introduce open-source tools interfacing directly with the popular puzzle application **SudokuPad** [17], facilitating both agentic tool-use interaction and standardized textual puzzle representations. The agentic interaction provides an API to fetch images of the current board state and access to all the annotation tools available in **SudokuPad** that human solvers usually rely on. Our textual format isolates logical reasoning from visual processing, enabling effective evaluation with current language models. Second, we present SUDOKU-BENCH, a carefully curated benchmark of 100 Sudoku variants, selected in collaboration with hosts from the *Cracking the Cryptic* YouTube channel. These puzzles span a wide range of difficulties and reasoning styles, deliberately chosen to test model performance across diverse logical pathways and puzzle-specific "break-ins."

Our experiments showcase SUDOKU-BENCH poses a striking challenge for current state-of-the-art models. Without tool assistance, even the strongest publicly available LLM evaluated solves fewer than 15% of the benchmark. Notably, most of the successful completions come from the simplest subset of $4 \times 4$ puzzles, with performance rapidly collapsing with larger and less conventional grids. This is observed in both the one-shot configuration (prompt a model to solve a puzzle in one response) and a multi-step configuration (multi-turn interaction between the model providing at least one digit and the user providing the updated board state).

Beyond benchmarking, Sudoku variants offer a fertile *laboratory* for reasoning research. An extensive, ever-growing supply of human-generated puzzles allows scalable difficulty progression, from simpler $4 \times 4$ puzzles suitable for small models to highly intricate $9 \times 9$ puzzles, the hardest of which can stump all but the best expert human solvers. Rich auxiliary data, including detailed expert solution transcripts and interaction traces, facilitate imitation learning. We include, as part of SUDOKU-BENCH thousands of hours of reasoning transcripts and actions taken when solving from *Cracking the Cryptic*, a popular YouTube channel dedicated to detailed demonstrations of solving Sudoku variants with over 250M views. This data is entirely available for researchers who wish to explore supervised approaches to learn and fine-tune models from human reasoning – qualitatively far beyond the depth and diversity of synthetic reasoning datasets with current state-of-the-art language models [11, 16].

The remainder of this paper proceeds as follows: Section 2 surveys Sudoku variants and their reasoning demands. Section 3 details the SUDOKU-BENCH dataset, text interface, and evaluation

framework. Section 4 presents baseline results and analyses of model failure modes. We review related work in Section 5, and conclude with open research directions in Section 6.

## 2 Background: Sudoku Variants

Traditional Sudoku involves completing a $9 \times 9$ grid such that each digit from 1 to 9 appears exactly once in every row, column, and $3 \times 3$ subgrid. This structure provides a foundation for numerous variants that introduce additional constraints. For instance, *Killer Sudoku* combines elements of Sudoku and Kakuro, requiring digits within outlined cages to sum to specified totals without repeats. *Thermometers* are paths of adjacent cells where digits must increase monotonically. Digits along *arrows* must sum to the digit in the circled cell at the base. *Kropki* dots between cells indicate specific relationships, such as consecutive numbers or a 1:2 ratio.

The availability of web-based puzzle-making tools allowed puzzle authors to invent their own variants. In early 2020, the puzzle-hosting site Logic Masters saw a surge in the number of puzzles posted. As of May 2025, more than 27,000 user-submitted variants are published on the site [1].

Puzzle creators frequently combine multiple constraints in unique ways. Often, these combined constraints result in puzzles starting with minimal or no digits, necessitating extensive logical reasoning to determine the initial placement, termed a "break-in." Such puzzles require solvers to meticulously explore the interaction of constraints, significantly diverging from the eager guessing often observed in reasoning LLMs (Section 4).

Beyond these standard constraint types, puzzle setters often employ meta-constraints, which involve deducing puzzle-specific parameters (e.g., "digits in a cage sum to an unknown value to be determined by solving," or "the line must be identified as either a palindrome or a renban sequence"). These meta-constraints add another layer of complexity and creative reasoning.

Puzzle authors are ultimately limited only by imagination, often developing whimsical and novel rulesets (e.g., puzzles themed around rats in mazes (Fig. 1)). Crucially, all Sudoku variants maintain a structured format: an $n \times n$ grid, natural-language puzzle rules, visual elements easily encoded as text, and a single unique solution. This structured yet flexible framework makes Sudoku variants exceptionally suitable for systematically investigating creative reasoning capabilities, meaning that the puzzles are very diverse and challenging but grounded and easy to verify if correct.

**Puzzle example: Ascension**   We illustrate some of these features with an example. Figure 2a highlights the novel interaction between a knight's move restriction and arrow constraints.

To find the puzzle's break-in, the solver must make three observations.

First, whatever the digit highlighted in green (`r4c6`, box 5), it must occur somewhere in box 2, but not in column 6 (by standard Sudoku rules), or along its arrow tip, or a knight's move away, thus can only occur in one of the two half-shaded cells `r1c4` or `r1c5`. This same pattern applies to the other cell groups highlighted by the other colors shown in the middle panel. The second observation is that since digits on the arrow must be smaller than the corresponding circled base, this creates a long-range chain dependency across the highlighted cells, namely, the circled cells shaded yellow, purple, green, blue, then red, must be monotonically increasing. This is a key insight but not enough to determine an exact digit yet.

The third observation is that the purple cell must be the sum of three Sudoku digits, the two in its arrow tip `r4c1` and `r4c2`, but one of which is equal to the yellow cell of `r7c3`, which itself is the sum of two Sudoku digits by arrow rules. The only digit that can be the sum of three Sudoku digits and leave enough room for the monotonic chain along green, blue and red, is six. Therefore `r4c6` must be six and the subsequent digits in the monotonic chain are forced (right panel).

In a video demonstrating this puzzle solve, an expert solver discovered this break-in in about 4.5 minutes, and a full puzzle solve taking about 35 minutes.[1] In all LLMs we tested, no model was able to make progress. For example, we show the reasoning summary of `Gemini 2.5 Pro Preview` (Fig. 2b), which was able to successfully parse and identify the puzzle constraints, but quickly resorts to guesswork and search. This highlights that there is still a gap between how LLMs reason and how humans prefer to reason; LLMs can rely on brute-force but humans will prefer to save time

---

[1] https://www.youtube.com/watch?v=-7OR_IK4Th8

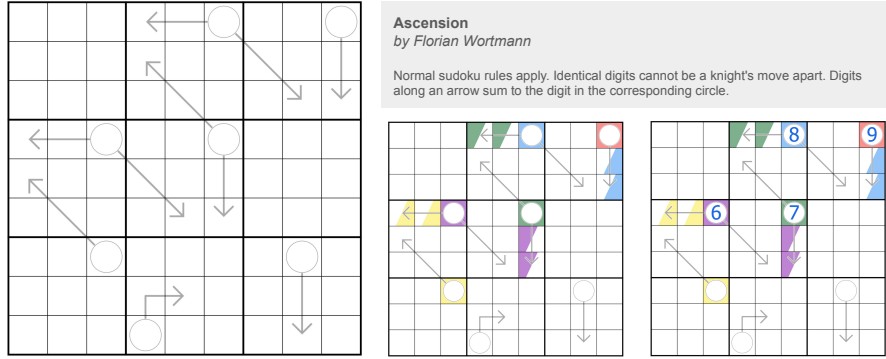

(a) Example of a logical break-in from the puzzle *Ascension*. Despite the initial grid being empty, the puzzle constraints collectively enforce a strict sequence of increasing digits from the lower left to the upper right shaded circles. Identifying and leveraging this subtle interplay forms the puzzle's essential insight. Though rated at an easy difficulty (2-star), current LLMs consistently fail to identify this logical entry point.

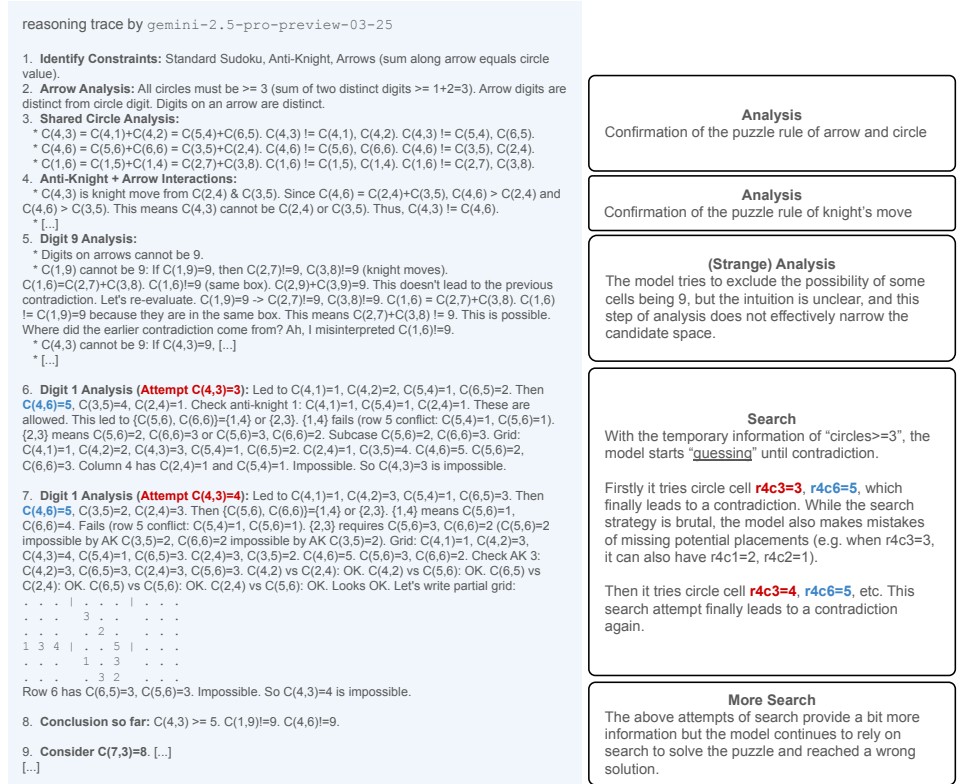

(b) `Gemini 2.5 Pro Preview`'s attempt to solve the puzzle *Ascension*. In contrast to the break-in by a human solver, the model failed to effectively narrow its search space and had to rely on a more brute-force search strategy, which did not lead to the correct solution.

Figure 2: *Ascension* example.

and energy by using precise logic to find shortcuts to correct digits. We hope to see this benchmark encouraging work on creating LLMs that reason in a more 'human-like' manner.

The *Ascension* example highlights two facets of Sudoku variants. First, although both knight-move and arrow constraints are commonplace, this specific interaction is unique to this particular puzzle. Therefore, the memorization-resistance of Sudoku variants is not exclusively due to the inclusion of novel rulesets; familiar constraints can induce a solving tactic never seen before. Indeed, some of the most difficult puzzles adopt deceptively simple rulesets. The second point is that for puzzles with few

or no given digits (as is common in variants), the search space is too large for initial guesswork to be effective. This also often necessitates a kind of meta-reasoning where one must decide at the outset what reasoning techniques should be applied, e.g. the use of coloring, set theory or looking at digit parity.

This pattern of needing to spend time at the beginning to understand how the constraints interact in a new manner is normal when humans tackle these puzzles. This also means that some of these initial deductions remain pertinent throughout the solve, meaning that in order to robustly solve some of these puzzles over 100s of steps will either require a form of memory, like a scratchpad, or a very long context window.

## 3 SUDOKU-BENCH: Dataset and Benchmark Design

We sought to select 100 puzzles that are representative of the breadth of Sudoku variants. To establish a graded evaluation curve, we selected 15 $4 \times 4$ puzzles, 15 $6 \times 6$ puzzles, and 70 $9 \times 9$ puzzles. The 15 $4 \times 4$ puzzles are included, in part, to measure progress in even modestly sized language models. Fifty of the $9 \times 9$ puzzles were curated by the hosts of *Cracking the Cryptic* exclusively for this benchmark. The selected puzzles evenly span difficulty ratings from novice-friendly "1-star" puzzles to expert-level "5-star" challenges that may require hours of careful analysis before any digits can be confidently placed. Twenty of the puzzles are difficult vanilla Sudokus, which were supplied by the puzzle company Nikoli, which popularized Sudoku in the 1980s. We aimed to create a smooth ramp in complexity such that an initial attempt at tackling the benchmark can yield some early success, but fully solving it will be vary challenging, and we hope that this benchmark will resist being solved for a significant time span.

**Text descriptions**   Each puzzle is given a pure text representation. For instance, Fig 3 shows a simple $4 \times 4$ puzzle whose line paths are represented as a sequence of `rxcy` (row x column y) coordinates, and the location of the dot is described as the two cells it lies between. The rules, visual elements, grid size, and initial board state (if any digits are given) are sufficient to unambiguously specify the puzzle and converted into a prompt.

While some of the most recent reasoning models have shifted toward multimodal inputs, we found that most, including OpenAI's o3 model, struggle in converting $9 \times 9$ puzzles into accurate coordinates. Puzzle benchmarks such as Enigma [27] and VGRP [23] emphasize the visual aspect of puzzles and require multimodal models. Given that current frontier models still struggle in exact specification of the visual elements of Sudoku puzzles, we opted to specify all elements precisely in text to isolate the creative reasoning process itself from visual understanding.

Each puzzle's text representation has been precomputed for puzzles on SUDOKU-BENCH. We provide the code for extracting text descriptions from a puzzle specified in SudokuPad, allowing researchers to utilize this harness in other puzzles.

Note that many of the puzzles would benefit from visual reasoning, some even potentially requiring it, since many of the break-ins are geometric and use symmetry, or have some rules that reference the shapes in the puzzle. Some puzzles can be very visually dense (See Bottom-Right in Fig 1) and current vision model we tested are not powerful enough to extract all the features, like the small numbers. We suspect that solving this benchmark using vision would represent a significant improvement over current multimodal LLMs.

### 3.1 Expert reasoning traces

A core question is whether advancing reasoning capabilities in LLMs can benefit from adopting more "human-like" thinking. In reinforcement learning models, pretraining on human supervision is common, while other work has shown that RL from scratch yields better performance in contained environments [25, 9, 14, 18]. Vanilla Sudoku is an interesting domain in that the strategies that humans use differ so significantly from search-based solvers [21], and this effect is especially pronounced in Sudoku variants.

The YouTube channel *Cracking the Cryptic* offers a particularly unique opportunity to explore the benefits of imitation learning. The channel contains over 3,000 published videos demonstrating the solving process of Sudoku variants. Notably, the hosts must verbally describe their thinking process,

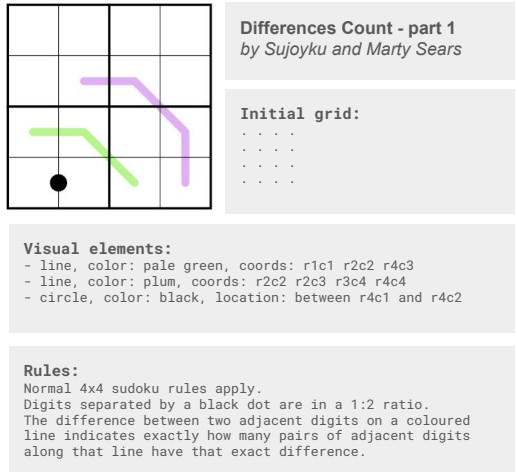

Figure 3: A text representation of a puzzle. The rules, initial grid, and a text description of visual elements are sufficient to unambiguously specify the puzzle.

explaining to the viewer each logical deduction. A typical puzzle takes the hosts around 60 minutes to solve, while some of the more difficult puzzles featured on the channel are over 3 hours in length. We developed a dataset consisting of the audio transcripts of each solve, together with a sequence of SudokuPad actions extracted from the video. The actions were extracted using a machine learning model trained on ground truth actions simulated on SudokuPad and then applied to video frames. This dataset is hosted on HuggingFace[2] under an MIT license in agreement with the hosts of the channel.

## 3.2 Dataset format

The SUDOKU-BENCH puzzle dataset[3] contains three subsets, `challenge_100`, `nikoli_100`, and `ctc`. The `challenge_100` is described above and represents the core benchmark. Additional puzzle data include `nikoli_100`, a collection of hand-made vanilla Sudokus supplied by Nikoli for this benchmark (20 of which are featured in `challenge_100`). The `nikoli_100` are designed to highlight creative or human-like reasoning in their solution paths, and may be applicable to many of the research approaches that use vanilla Sudoku as a testbed (Section 5). The `ctc` includes 2,565 Sudoku variants that have been solved on *Cracking the Cryptic*. Due to the breadth and variety of Sudoku variants, the text representation of each puzzle in `ctc` has not undergone manual checking, and an unambiguous representation of the board would require a screenshot in some cases.

## 3.3 SudokuPad environment

We also provide tools for interacting with SudokuPad in an agentic environment. SudokuPad enables common note-taking strategies used by human solvers, including color-coding cells (as in Fig. 2a) or providing candidate digits or pencil marks to cells. Our simple harness allows models to directly interface with the application to make use of these tools. Using SudokuPad in-the-loop may fit well with related benchmarks that evaluate reasoning models (including vision language models) in simple game environments [19, 23]. Our evaluation in this paper (Section 4) uses text interaction (relying only on SudokuPad for the initial puzzle data extraction). We make all of these SudokuPad tools available for researchers on our repository `https://github.com/SakanaAI/Sudoku-Bench`.

## 3.4 Evaluation Framework

**Multi-step and single-shot**   We evaluate models in both multi-round and single-shot configurations. In a multi-round setup, we prompt the model to analyze the board and give at least one valid digit

---

[2]`huggingface.co/datasets/SakanaAI/Sudoku-CTC-Reasoning`
[3]`huggingface.co/datasets/SakanaAI/Sudoku-Bench`

placement per response. We clarify that this is a committed digit(s) that cannot be undone (in the model's reasoning trace, any amount of internal backtracking is possible in order to deduce the digit). Once the digit is placed, the user displays the updated board state. We continue until the puzzle is solved or the LLM misplaces any digit. In the multi-round setting, we track both the solve rate and correct digit placements per puzzle. To keep the context window manageable, we keep the most recent 5 responses from the LLM in context, while always keeping the first user message with the puzzle specification and instructions. We report the averages as **average solve rate** and **average correct digits**. In our evaluation, we run a single evaluation per model and per puzzle, so the average is across the 100 puzzles in the set.

In the single-shot configuration, we prompt the model to provide a solution in a single response. A single-shot configuration is appropriate for evaluating models with sufficiently large context, or for a more straightforward evaluation of the smaller $4 \times 4$ puzzles. In the single-shot setting, we report only the **average solve rate**.

# 4   Baseline Performance and Analysis

We evaluated the current generation of state-of-the-art large language models on SUDOKU-BENCH, revealing substantial difficulty posed by these Sudoku variants. Table 1 summarizes model performance across puzzle sizes and interaction modes on benchmark. Even leading models such as o3 mini high and Gemini 2.5 pro preview demonstrated solve rates below 15% for the complete set. Notably, performance varied significantly by puzzle size: models generally solved smaller $4 \times 4$ puzzles at rates between 40% to 73%, but performance sharply declined for $6 \times 6$ grids and dropped nearly to zero on $9 \times 9$ puzzles, underscoring the rapid escalation in complexity.

Comparing single-shot to multi-step evaluation modes, allowing iterative feedback slightly improved outcomes for smaller puzzles but did not meaningfully impact results for larger puzzles. The minimal difference between modes suggests that the fundamental difficulty for these models lies not merely in incremental reasoning but in effectively identifying initial logical breakthroughs.

| Model | Multi-step correct placements | | | | Multi-step solve rate (%) | | | | Single-shot solve rate (%) | | | |
|---|---|---|---|---|---|---|---|---|---|---|---|---|
| | 4×4 | 6×6 | 9×9 | **All** | 4×4 | 6×6 | 9×9 | **All** | 4×4 | 6×6 | 9×9 | **All** |
| O3 Mini High | 9.7 | 0.7 | – | **–** | 60.0 | 0.0 | – | **–** | 73.3 | 6.7 | 2.9 | **14.0** |
| Gemini 2.5 Pro | 11.6 | 0.6 | 1.8 | **3.1** | 73.3 | 0.0 | 0.0 | **11.0** | 60.0 | 13.3 | 0.0 | **11.0** |
| Qwen 3.235B A22B | 6.5 | 1.1 | 0.7 | **1.7** | 40.0 | 0.0 | 0.0 | **6.0** | 53.3 | 0.0 | 0.0 | **8.0** |
| Qwen 3.30B A3B | 1.3 | 0.0 | 0.3 | **0.4** | 6.7 | 0.0 | 0.0 | **1.0** | 46.7 | 0.0 | 0.0 | **7.0** |
| DeepSeek R1 | 9.5 | 0.8 | 1.1 | **2.3** | 60.0 | 0.0 | 0.0 | **9.0** | 40.0 | 0.0 | 0.0 | **6.0** |
| Grok 3 Mini | 8.5 | 0.7 | 0.9 | **2.0** | 53.3 | 0.0 | 0.0 | **8.0** | 40.0 | 0.0 | 0.0 | **6.0** |
| Qwen QwQ 32B | 5.0 | 0.7 | 0.6 | **1.3** | 26.7 | 0.0 | 0.0 | **4.0** | 40.0 | 0.0 | 0.0 | **6.0** |
| Qwen 3 32B | 4.3 | 0.5 | 0.5 | **1.0** | 26.7 | 0.0 | 0.0 | **4.0** | 40.0 | 0.0 | 0.0 | **6.0** |
| Claude 3.7 Sonnet (Thinking) | 8.1 | 1.1 | – | **–** | 40.0 | 0.0 | – | **–** | 33.3 | 0.0 | 0.0 | **5.0** |
| GPT 4.1 | 2.3 | 0.2 | 0.3 | **0.6** | 13.3 | 0.0 | 0.0 | **2.0** | 13.3 | 0.0 | 0.0 | **2.0** |
| Gemini 2.0 Flash | 0.5 | 0.1 | 0.2 | **0.2** | 0.0 | 0.0 | 0.0 | **0.0** | 0.0 | 0.0 | 0.0 | **0.0** |
| Gemma 3 27B IT | 0.1 | 0.1 | 0.5 | **0.3** | 0.0 | 0.0 | 0.0 | **0.0** | 0.0 | 0.0 | 0.0 | **0.0** |
| Llama 4 Maverick | 0.2 | 0.5 | 0.4 | **0.4** | 0.0 | 0.0 | 0.0 | **0.0** | 0.0 | 0.0 | 0.0 | **0.0** |

Table 1: **Sudoku-Bench leaderboard.** Performance comparison of various LLMs on Sudoku-Bench. Percentage of puzzles completely solved for each evaluation mode (multi-step vs. single-shot), stratified by grid size. The right-most **All** columns aggregate across grid sizes (15 puzzles for 4×4 and 6×6, 70 for 9×9). In the multi-step setting, a model is prompted to provide any number of digits in its response, with the user providing an updated board state at each turn. Interaction is terminated if the model makes an incorrect placement. The average number of correct placements are presented in the first column set. In the single-shot setting the model is prompted to solve the entire puzzle in a single response. "–" indicates that fewer than the required number of responses were available due to cost limitations, so an aggregate could not be computed.

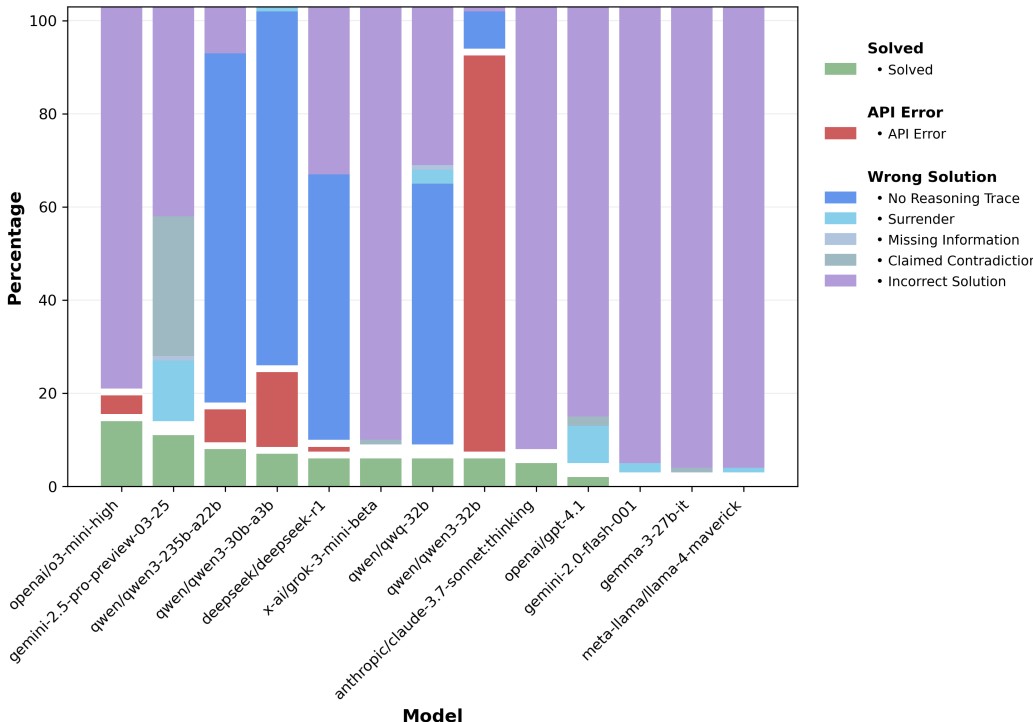

Figure 4: Response categorization for the single-shot setting.

**Categorizing model failures** Analyzing model failures indicated several recurring patterns which we categorize in Fig. 4. The most common failure mode was presenting with confidence an *Incorrect Solution*. Other failure modes included *Surrender* (model explicitly gives up), *Missing Information* (model incorrectly claims puzzle information or given constraints are incomplete), and *Claimed Contradiction* (model mistakenly identifies contradictions in the puzzle rules). Of note is *Missing Information*. Since variants are not as densely represented in the training set of foundation models compared to vanilla Sudoku, it appears the new rules and variants throw them off, most notably due to the fact that variants typically have fewer starting digits (often none) compared to the minimum of 17 in a vanilla $9 \times 9$ Sudoku. In addition, a part of model responses contain *No Reasoning Trace* so we cannot make a fine-grained categorization of its error type, otherwise we use `Claude-3.5-Haiku` to classify a wrong solution response into one of the other four error types.

## 5 Related Work

SUDOKU-BENCH complements existing benchmarks designed to evaluate advanced reasoning in artificial intelligence, with a particular focus on Sudoku variants as a structured domain for assessing creative and logical deduction.

**Benchmarks targeting creative deductive insight** Benchmarks such as the Abstraction and Reasoning Corpus (ARC; 3) present diverse tasks to test reasoning and generalization beyond pattern memorization. SUDOKU-BENCH similarly introduces novel constraints for each puzzle, resisting memorization through a continuous influx of unique puzzles. Unlike ARC, which emphasizes tasks simple for humans but challenging for AI, Sudoku variants span a broader difficulty spectrum, including puzzles challenging even for expert human solvers. Nonetheless, Sudoku puzzles offer recognizable logical breakthroughs readily appreciated by human novices, making SUDOKU-BENCH a valuable resource for precise evaluation of creative reasoning.

**Puzzle-centric reasoning datasets** Several benchmarks focus on puzzle-solving for evaluating reasoning skills [5]. For instance, PUZZLES [4] compiles canonical logic puzzles; Tyagi et al. [26]

systematically analyze grid puzzle-solving by LLMs; and ENIGMAEVAL [27] evaluates a large suite of problems from puzzle competitions. Recent additions include VGRP-BENCH [23] for visual-grid reasoning, LOGICGAME [7] for rule-based reasoning, and PUZZLEPLEX [13] for evaluating conversational agents' reasoning. BALROG [19] evaluates LLM and VLM reasoning in complex game environments and could be extended using tools from SUDOKU-BENCHto include SudokuPad as an environment.

**Sudoku as a reasoning testbed**  The standard Sudoku puzzle has been extensively utilized in machine learning research. Models include Recurrent Relational Networks [20] employing message-passing, differentiable SATNet consistency layers [28], masked-denoising and diffusion methods [10, 30], and Kuramoto-inspired oscillator dynamics [15]. Further, large language models have achieved human-level accuracy through structured prompting and reasoning decomposition [12]. [24] showed a high solve rate on vanilla Sudokus by training on a sequence of steps from a solver. SUDOKU-BENCH extends this research tradition by incorporating diverse and novel puzzle constraints, enabling evaluations that specifically target multi-step, strategic, and creative reasoning.

# 6   Discussion

**The role of tool use**  Evaluating model reasoning can be distinguished by whether external tools, such as constraint solvers or code execution environments, are available. Without tool use, the evaluation specifically assesses the model's intrinsic reasoning capabilities, including logical deduction, maintaining global consistency, and internally generating creative insights, akin to solving puzzles by hand. This approach emphasizes pure cognitive reasoning skills and has been the primary evaluation mode presented in our baselines (Section 4).

Conversely, allowing tool use tests the model's ability to translate a given puzzle into a formal representation suitable for external solvers, effectively interact with these tools, and interpret solver results correctly. Standard Sudoku puzzles become straightforward when a solver is employed. Variants that only employ standard constraints such as arrows, cages, etc, are also easily solved by code execution. A third category of puzzles require natural language understanding and are not straightforward to interpret as a constraint satisfaction problem. This third category is itself a meaningful test for reasoning models with tool-use enabled. However, our current intention is to assess the reasoning required to find a puzzle's "break-in," and many puzzles such as *Ascension* from Fig. 2a are easily solved by tool-use, but the solution path would be substantially different than that intended by the puzzle setter. Therefore we selected the 100 puzzles of SUDOKU-BENCH for evaluating models without tool-use. Future work could consider a separate tool-use track, potentially with a different collection of puzzles.

**Limitations**  The current evaluation results are limited to text interfaced models. We would like to incorporate evaluation of VLMs in the future when they are capable of reading puzzles.

**Societal impact**  The release of SUDOKU-BENCH provides a platform for assessing large language models on difficult Sudoku variant puzzles that challenge even experienced human solvers. Our evaluation results, consistent with findings from previous research, demonstrate that LLMs employ fundamentally different solving strategies compared to human approaches. As LLMs continue to advance in capability, we anticipate a future where human puzzle creators can learn from and incorporate AI-discovered strategies to develop even more intriguing and sophisticated variants, creating a synergy between human and artificial intelligence.

**Conclusion**  We introduced SUDOKU-BENCH, a unified benchmark built around modern Sudoku variants that systematically stress long-horizon deduction, rule-interpretation, and strategic planning. In addition, the benchmark is uniquely suited for evaluating creative reasoning via the rich and varied collection of break-ins featured in most puzzles. The benchmark includes a curated puzzle corpora with textual representations, providing a controlled substrate for measuring how well language models cope with novel, tightly coupled constraints. Baseline experiments show that frontier LLMs solve fewer than 15% of instances without external tools, and performance falls sharply on $9 \times 9$ variants—evidence that substantial headroom remains for improvements.

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

## A  Appendix

We show a $6 \times 6$ puzzle (*Sumthings*) sovled by `Gemini 2.5 Pro Preview` in Fig 5. The model adopted a similar strategy as in the *Ascension* puzzle. In this puzzle, the model successfully reduces the search space to a reasonably small size and uses search to find the correct solution. However, as demonstrated by the previous example of *Ascension*, such strategy becomes ineffective when the puzzle complexity increases and one has to rely on "break-in" techniques to effectively reduce the search space.

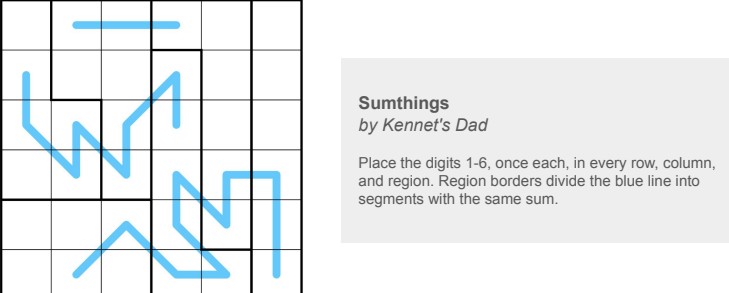

(a) Description of the puzzle *Sumthings*.

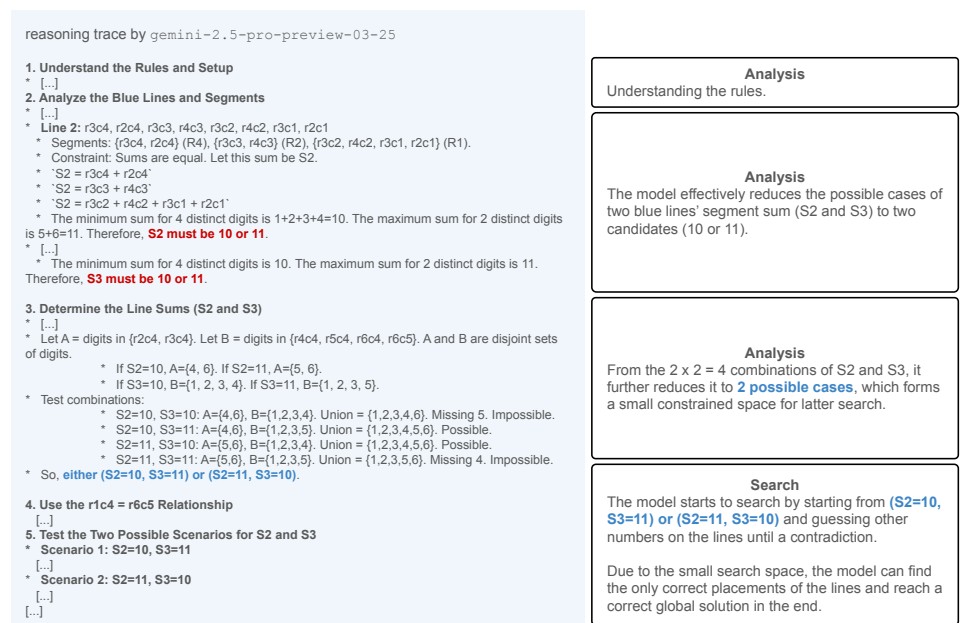

(b) Gemini 2.5 Pro's solution to the puzzle *Sumthings*.

Figure 5: *Sumthings* example.

