# OpenReview forum: "Sudoku-Bench: Evaluating creative reasoning with Sudoku variants"
_NeurIPS.cc/2025/Datasets_and_Benchmarks_Track — Submitted to NeurIPS 2025 Datasets and Benchmarks Track_

### Official Review · Reviewer_PgRj · 2025-06-26

**Rating:** 4
**Confidence:** 3

**Summary:**

This paper introduces SUDOKU-BENCH, a new benchmark designed to evaluate the creative, multi-step logical reasoning capabilities of large language models (LLMs) using diverse Sudoku variants. The authors highlight that existing benchmarks often reward memorization, whereas Sudoku variants necessitate novel logical deductions or "break-ins" due to their unique and often subtly interacting constraints. The benchmark includes a curated set of 100 puzzles with standardized text-based representations and provides flexible tools compatible with publicly available puzzles. It also incorporates thousands of hours of expert human reasoning traces from the "Cracking the Cryptic" YouTube channel, offering rich auxiliary data for imitation learning. Baseline experiments indicate that current state-of-the-art LLMs solve fewer than 15% of the puzzles unaided, with performance sharply declining on larger and less conventional grids, suggesting considerable room for improvement in LLM reasoning capabilities.

**Dataset Code Accessibility:**

Yes

**Dataset Code Comments:**

The dataset has been released.

**Ethical Considerations:**

No, there are no or only very minor ethics concerns

**Final Justification:**

My concerns are addressed so I maintain the original positive rating.

**Limitations Weaknesses:**

- **Exclusion of Visual Reasoning**: The current evaluation is explicitly limited to text-interfaced models to isolate creative reasoning from visual understanding. However, the paper acknowledges that many puzzles, especially those with geometric break-ins or visually dense elements, could benefit from or even require visual reasoning. This design choice, while justified for isolating reasoning, means the benchmark does not fully assess the multimodal reasoning capabilities of LLMs for this type of problem, which could be a relevant area for future work.

- **Context Window Management in Multi-step Evaluation**: In the multi-round setup, the context window is limited to the most recent 5 responses from the LLM, in addition to the initial puzzle specification. While this helps manage context, it could potentially hinder models that require a longer memory or more extensive internal scratchpad for complex, long-horizon puzzles, as suggested by the paper itself. It would be beneficial to discuss the potential impact of this limitation more deeply or explore alternative context management strategies.

- **Unchecked Text Representation for ctc Dataset**: The paper states that the text representation for puzzles in the ctc dataset (2,565 Sudoku variants) has not undergone manual checking, and some may require screenshots for unambiguous representation. While challenge_100 is the core benchmark, the potential ambiguity in the larger ctc dataset could be a limitation for researchers wishing to utilize that broader collection without additional validation effort.

- **Definition of "Creative Reasoning"**: While the paper posits that Sudoku variants evaluate "creative reasoning," the definition of this term could be further elaborated or operationalized within the context of LLM capabilities. The paper describes it in terms of "novel logical breakthroughs" or "break-ins", but a more explicit connection to broader definitions of creativity in AI research could strengthen this claim.

**Strengths Contributions:**

- **Novelty and Problem Space**: The paper addresses a pertinent limitation of current LLM reasoning benchmarks by focusing on Sudoku variants that resist memorization and require novel logical insights. The domain is well-suited for reasoning research due to its diverse, challenging, yet structured nature. The concept of "break-ins" for puzzles with minimal initial digits is a well-articulated challenge for LLMs.

- **Comprehensive Benchmark Design**: SUDOKU-BENCH is a carefully curated dataset of 100 puzzles, spanning a range of difficulties and grid sizes (4×4, 6×6, 9×9), including custom puzzles from "Cracking the Cryptic" and traditional Sudokus from Nikoli. This graded difficulty allows for a comprehensive evaluation curve.

- **Standardized Text Representation and Tools**: The provision of a standardized text-based puzzle representation effectively isolates logical reasoning from visual processing, which is valuable for evaluating current LLMs that may struggle with multimodal inputs for complex puzzles. The open-source tools for interfacing with SudokuPad and extracting text descriptions are a useful contribution for researchers.

- **Rich Auxiliary Data for Research**: The inclusion of expert human reasoning traces, derived from "Cracking the Cryptic" YouTube videos, provides a unique and extensive dataset for exploring human-like reasoning and imitation learning in LLMs. This distinguishes it from many synthetic reasoning datasets.

- **Thorough Baseline Evaluation**: The paper presents a detailed baseline performance analysis of various state-of-the-art LLMs in both multi-step and single-shot configurations, demonstrating the current limitations of these models on the benchmark. The categorization of model failure modes (e.g., Incorrect Solution, Surrender, Missing Information) provides insightful analysis into LLM shortcomings in this domain.

---

> ### Author Rebuttal · Authors · 2025-07-30
>
> We thank Reviewer PgRj for their positive and comprehensive summary of our work. We are grateful that the reviewer found our benchmark to be well-motivated, the dataset significant, and the evaluation extensive. We believe we can fully address the remaining excellent points and have clarified them below.
>
> **W1 (Exclusion of Visual Reasoning): We appreciate the reviewer understanding our rationale for excluding visual reasoning to isolate the logical challenges, which remains an open problem for future multimodal research.**
>
> We thank the reviewer for recognizing our rationale. As we state in the paper (L111-L119), this was a deliberate choice to isolate logical reasoning from the current limitations of visual perception in VLMs. We fully agree that incorporating multimodal reasoning is an important future direction and explicitly present this as an open challenge posed by our benchmark.
>
> **W2 (Context Window Management): The limited context window was a practical choice for fair evaluation, but we clarify that the model always has access to the full problem description and current board state, which is sufficient for logical deduction.**
>
> This is an insightful point. The decision to limit the conversational history to the 5 most recent turns was a practical one for tractability and fair comparison. However, we stress that the original prompt, the full puzzle description, and the current board state are always in context. Thus, the model has all necessary information to progress. The recent turns allow it to carry over unstated insights, but any logical step can be re-derived from the primary information if needed. We acknowledge this choice could be a limitation for puzzles requiring memory over hundreds of steps and, as suggested, will add a sentence to the Discussion to address this and encourage future work on more sophisticated memory mechanisms.
>
> **W3 (Unchecked CTC Dataset): We clarify that the large ctc dataset is an auxiliary training resource, distinct from the manually curated challenge\_100 evaluation benchmark. Its value lies in providing training data for puzzles whose visual complexity defies simple text encoding.**
>
> We appreciate the reviewer highlighting this. To clarify, the core challenge\_100 benchmark was manually curated and checked. The larger ctc dataset is provided as an auxiliary resource for training. An important point we will now emphasize is that many of these sudoku variants, with their whimsical visual styles (e.g., Rat Run in Fig 1), are nearly impossible to faithfully encode into text automatically. SudokuPad gives authors immense freedom with visual elements, creating many edge cases for automated text extraction. We provide this data as a valuable resource for the community, especially for training VLMs, as a screenshot and ruleset still define a valid puzzle solvable by humans.
>
> **W4 (Definition of "Creative Reasoning"): Acting on this excellent suggestion, we have strengthened the paper by incorporating established definitions of "Insight" from cognitive psychology, which more precisely defines the reasoning our benchmark targets.**
>
> This is a valuable suggestion. We conducted a literature review on “Insight” or “Aha” moments as studied in Cognitive Science. Research pioneered by Gestalt psychologists identifies Insight as the result of a transition from associative thinking (pattern matching) to “productive thinking”--a novel understanding of the problem's deep structure. Many argue this capacity is definitive of human reasoning (Vitello & Salvi, J Intell. 2023). By leveraging this rich field, we have updated the paper (primarily in the Discussion) to reflect these more precise definitions. We thank the reviewer for this suggestion, which has significantly strengthened the paper's framing.

---

> > ### Comment · Reviewer_PgRj · 2025-08-04
> >
> > Thanks for your response. My concerns have been addressed. I will maintain my positive rating.

---

### Official Review · Reviewer_Cch1 · 2025-07-01

**Rating:** 4
**Confidence:** 3

**Summary:**

This paper introduces SUDOKU-BENCH, a benchmark designed to evaluate the creative and multi-step logical reasoning capabilities of large language models (LLMs) using challenging Sudoku variants. The benchmark includes 100 carefully curated puzzles of varying sizes (4×4, 6×6, 9×9), with text-based representations and tools for interaction via SudokuPad.

**Dataset Code Accessibility:**

Yes

**Ethical Considerations:**

No, there are no or only very minor ethics concerns

**Limitations Weaknesses:**

1. While the authors acknowledge that visual reasoning is central to solving some variants, current baselines only evaluate text-based interfaces. This omits an important aspect of human-like reasoning.
2. All evaluations use general-purpose LLMs. It remains unclear whether models fine-tuned on Sudoku strategies (or trained via imitation learning on the included expert traces) could perform substantially better.
3. The “solve rate” metric only rewards exact completions, which might not fully reflect partial understanding. Intermediate progress (correct placements, sub-goal detection) could provide more nuanced signal.

**Strengths Contributions:**

1. Unlike many benchmarks that reward memorization, SUDOKU-BENCH uses hand-crafted Sudoku variants that require multi-step logical insights (“break-ins”) and resist template-based recall .
2. Sudoku variants provide a constrained but expressive testbed for evaluating reasoning, offering verifiable solutions, scalable difficulty, and clear logical objectives—making them well-suited for studying long-term planning and symbolic abstraction.

---

> ### Author Rebuttal · Authors · 2025-07-30
>
> We thank the reviewer for their thoughtful review and insightful comments.
>
> **W1 (Visual Reasoning): We deliberately focused on text to isolate logical reasoning from current VLM perception failures. We provide a concrete example of a state-of-the-art VLM failing to parse even a simple 6x6 puzzle grid, justifying our approach.**
>
> We completely agree that visual reasoning is an important direction. This was a deliberate design choice, discussed in Section 3 (L145-L150, L154-L159). Our initial approach was indeed to highlight the visual aspect, but we found that current VLMs struggle to accurately parse the dense, precise information in these grids. For example, when we prompted a frontier VLM (o3-pro) to convert Figure 5 into a text specification, it failed to extract the exact path of the blue lines and the coordinates of the irregular regions. Even with favorable prompting, it failed to parse the grid correctly. If a VLM cannot even identify the ruleset—where any off-by-one error can invalidate the puzzle—it cannot solve it. Given that Figure 5 is a relatively simple 6x6 grid, this demonstrates a critical bottleneck. We therefore focused on high-quality text-only descriptions to study creative reasoning without the confound of VLM perception limitations. This also allowed us to assess a wider range of models. To support future visual reasoning research, our SudokuPad tools facilitate easy screenshot extraction (L194). We have updated the paper to expand on this rationale.
>
> **W2 (Fine-tuned Models): We agree fine-tuning is a key future direction; our benchmark and the large-scale human-trace dataset were specifically designed to enable this line of research.**
>
> This is an excellent point. Our focus here is to introduce the benchmark and establish its difficulty for general-purpose foundation models. We are pursuing our own fine-tuning experiments but do not have results ready for this submission. However, we included the large-scale dataset of expert reasoning traces from Cracking the Cryptic (Section 3.1, L123-129) precisely to enable and encourage future research into imitation learning and fine-tuning (L42, L129). We believe this is a very promising direction and look forward to seeing work from the community.
>
> **W3 (Granularity of "Solve Rate"): Our evaluation framework already includes a more granular, partial-credit metric: the average number of correctly placed digits**
>
> While we report the overall solve rate, we also provide a more granular metric as suggested. As detailed in our Evaluation Framework (Section 3.4, L150), for the multi-step evaluation, we track and report the average correct digits placed per puzzle. This metric provides a partial-credit signal that reflects intermediate progress. This quantitative measure is complemented by our qualitative categorization of failure modes (Figure 4), which offers further nuance.

---

### Official Review · Reviewer_XjGR · 2025-07-02

**Rating:** 4
**Confidence:** 4

**Summary:**

SUDOKU-BENCH is a reasoning benchmark built from tough, unconventional Sudoku variants. Each puzzle layers unique or subtly interacting constraints, providing a compact yet diverse test bed for multi-step, creative reasoning. Initial results show cutting-edge language models solve under 15 % of these puzzles without help, revealing ample headroom for improving long-horizon, strategic reasoning in LLMs.

**Additional Feedback:**

Please see the weakness part.

**Dataset Code Accessibility:**

Yes

**Dataset Code Comments:**

The data has already been uploaded to huggingface.

**Ethical Considerations:**

No, there are no or only very minor ethics concerns

**Final Justification:**

I maintain my score, as the limitation of the dataset, covering only a limited range of reasoning patterns, cannot be resolved without incorporating additional types of reasoning data.

**Limitations Weaknesses:**

1. The benchmark is synthetic and heavily relies on some specific skills and patterns, which may not align with the requirements of daily tasks.
2. The benchmark lacks diversity, as the task set is limited and exclusively consists of Sudoku variants.
3. The authors need to clarify what type of reasoning is essential for these Sudoku tasks, and which kind of patterns is targeted for this benchmark.

**Strengths Contributions:**

1. The authors introduce a SUDOKU benchmark, framing long-horizon reasoning around unconventional Sudoku variants.
2. The puzzle set is explicitly selected for diversity, escalating difficulty, and minimal susceptibility to memorization. The standardized text representation and tooling make each instance unambiguous and machine-friendly, fostering consistent comparison across systems.
3. The authors show leading LLMs solve <15 % of puzzles without assistance, quantifying the current gap between state-of-the-art and human-level play. This stark performance gap both validates the benchmark’s difficulty and highlights concrete opportunity for advancing multi-step, strategic reasoning.

---

> ### Author Rebuttal · Authors · 2025-07-30
>
> We thank the reviewer for their feedback.
>
> **W1 (Benchmark Relevance): The benchmark's hand-crafted puzzles test core components of general-purpose reasoning. We have strengthened this argument by incorporating established literature on "Insight" from cognitive psychology**
>
> One of the benchmark's core strengths is that puzzles are hand-crafted by human experts to challenge human intellect. We argue that the skills required are core components of general-purpose reasoning, and a model capable of solving these diverse puzzles would possess robust and generalizable abilities. As suggested by Reviewer PgRj, we have added discussions around the concept of “Insight” from Cognitive Psychology. Proponents argue that these “aha” moments are fundamental to human thinking (Vitello & Salvi, J Intell. 2023), as it exemplifies an almost co-creative or “productive” stance with respect to difficult problems where solution paths are not immediately obvious. We hope this additional discussion clarifies why we think this benchmark is unique and of particular relevance to the AI reasoning research community.
>
> **W2 (Benchmark Diversity): The benchmark's diversity stems from its wide array of logical rules and constraints, not the shared Sudoku grid format, which provides a controlled evaluation framework.**
>
> While all tasks use a Sudoku grid, the benchmark's diversity comes from the rules and constraints, which vary dramatically. As described in Section 2, this includes a wide array of constraint types (e.g., Killer cages, thermometers, arrows, Kropki dots), novel combinations thereof, and whimsical, natural-language-only rulesets. This "structured yet flexible framework" (L87) is a key strength, allowing for immense logical diversity within a controlled and verifiable environment, while resisting memorization.
>
> **W3 (Reasoning Type): We have clarified the specific type of reasoning targeted, both by pointing to existing explanations and by adding a new discussion on "Insight" in problem-solving.**
>
> Section 2 ("Background: Sudoku Variants") and the detailed Ascension puzzle walkthrough (Figure 2) are dedicated to explaining the specific type of reasoning targeted. Furthermore, our new discussion of "Insight" from Cognitive Psychology, mentioned above, offers additional clarity on this. We thank the reviewer for pressing on these points, as the resulting discussion significantly improves the paper's clarity.

---

### Official Review · Reviewer_zJf8 · 2025-07-03

**Rating:** 3
**Confidence:** 4

**Summary:**

The SUDOKU-BENCH paper introduces a novel benchmark designed to rigorously evaluate the creative, multi-step logical reasoning capabilities of large language models (LLMs) using challenging Sudoku variants. The authors curate a balanced set of 100 puzzles (15 4×4, 15 6×6, and 70 9×9), each accompanied by a standardized text-based description (rules, grid state, and visual elements), and provide both a single-shot and multi-round evaluation framework. To facilitate “human-like” reasoning, they release tools for interacting with SudokuPad and a rich corpus of expert solution transcripts sourced from the Cracking the Cryptic YouTube channel. Baseline experiments with 17 state-of-the-art LLMs reveal solve rates below 15% overall—dropping near zero on 9×9 variants—and highlight recurring failure modes such as brute-force guessing, premature surrender, and missing constraint interpretation

**Additional Feedback:**

It would be great to see how different prompting strategies, like Tree-of-Thought affect the performance.

**Dataset Code Accessibility:**

Yes

**Ethical Considerations:**

No, there are no or only very minor ethics concerns

**Final Justification:**

I’m finding it difficult to justify an increase in the score. While I believe the benchmark is a valid contribution, the paper lacks sufficient depth—and at times breadth—in several key aspects of benchmark construction, experiments and analysis. The rebuttal didn’t substantially address these aspects either.

**Limitations Weaknesses:**

W1. Error analysis is very similar to non-creative reasoning benchmarks like math or logical reasoning. It does not really reflect or indicate the difficulty of Sudoku-bench. Also how many examples in total were inspected to draw the conclusion? Do the examples selected have enough coverage of the characteristics of the entire dataset?
W2. No case studies about the solution were presented.
W3. No categorization of the problems is presented for the entire dataset.

**Strengths Contributions:**

S1. Memorization-resistant domain: Sudoku variants—with novel or subtly interacting constraints—demand genuine insight rather than pattern recall.

S2. Diverse, graded difficulty: Inclusion of small (4×4) through expert-level (9×9) puzzles enables progress tracking across a smooth difficulty curve.

S3. Holistic evaluation framework: Single-shot and multi-round protocols, along with fine-grained failure categorization, offer deep insights into how and where models falter.

---

> ### Author Rebuttal · Authors · 2025-07-30
>
> We thank the reviewer for their detailed feedback and constructive suggestions.
>
> **W1 (Error Analysis). The error analysis is comprehensive, and its categories capture complex logical failures unique to our benchmark.**
>
> We have clarified in the paper that while the error categories in Figure 4 have general names, their root cause in Sudoku-Bench is distinct from typical reasoning benchmarks. For instance, an 'Incorrect Solution' in our context rarely stems from a simple arithmetic or rule-following error. As our 'Ascension' case study shows (Section 2, Figure 2), it almost always results from a fundamental failure to deduce a non-obvious logical pathway (a 'break-in'), a complex failure mode our benchmark is specifically designed to elicit. The analysis in Figure 4 was performed on all attempts across all baseline models for the 100-puzzle benchmark, ensuring full coverage of the characteristics observed in our experiments.
>
> **W2 (Case Studies) We have added more qualitative case studies to the appendix, including a new analysis of a Gemini 2.5 Pro failure mode as requested.**
>
> Given the reviewer's interest in providing specific case studies about the puzzles' solutions, we added additional qualitative examples (beyond Figures 2 and 5) in the appendix of our paper.
> In case the reviewer deems it necessary, we would also be open to analyzing and comparing the LLM solution with our collected human reasoning data for future revisions of our work.
> In particular, we add a figure similar in style to Figure 5, from Gemini 2.5 Pro from the puzzle Arrowsome:
>
> ```
> 2.  Analysis of Region 3 (r3c1-r4c3), which contains cells from both A1 ({r3c1, r4c2}) and A2 ({r3c3, r4c3}), reveals that for these four cells to be distinct Sudoku digits within the region, one arrow (A1 or A2) must use {1,1,4} and the other must use {1,2,3}. Specifically, the sets of digits must be {1,4} and {2,3}.
> 3.  This forces the 'external' digits on these arrows to be 1 (r2c1=1 and r5c4=1).
> 4.  Analysis of Region 6 (r5c4-r6c6) and Arrow A3 forces r6c5=6 and forces A3 to use the digits {1,2,3}, which implies r4c5=1 and {r5c5, r6c6}={2,3}.
> 5.  The combination {r3c1,r4c2}={1,4} and r4c5=1 leads to a contradiction:
>     * If r4c2=1, then Row 4 has two 1s (r4c2 and r4c5), violating Sudoku rules.
>     * Therefore, r4c2=4, which forces r3c1=1.
>     * But now Column 1 has r2c1=1 and r3c1=1, violating Sudoku rules.
> ```
>
> The failure comes from the fact that while the LLM is correct in step 2 that the digit sets must be {$*1,4*$, $*2,3*$}, it only explores the case where A1={1,4} and A2={2,3}, and not the other valid assignment. This leads it to incorrectly conclude the puzzle is self-contradictory. We believe this showcases an interesting failure mode and thank the reviewer for this suggestion. We would also be open to analyzing LLM solutions against our collected human reasoning data in future work.
>
> **W3 (Problem Categorization). We will enhance puzzle categorization by adding objective metadata (e.g., human difficulty ratings, solve times) to supplement our existing discussion of puzzle diversity**
>
> The puzzles were intentionally selected to cover a wide spectrum of types and difficulties, as described in Section 3. To further aid researchers, we will add available metadata for the puzzles in the camera-ready version, such as difficulty ratings from Logic Masters Germany and original human solve times from Cracking the Cryptic, where applicable. We believe this will offer a finer granularity of difficulty compared to categorizing only by grid size. While we discuss puzzle categorization in Section 3 (L129-139), we hope this additional table will make it more explicit.
>
> **Additional Feedback (Prompting):**
>
> As suggested, we have added new results to the Appendix using majority vote/self-consistency evaluation. To address the reviewer's suggestion, we have collected results with different prompting and evaluation strategies. The table below shows results for single-step generation (n\_samples=5, T=0.6) using majority vote on the subset of the benchmark.
>
> | Model | Correct Placements (4x4) | Solve Rate (%) (4x4) | Correct Placements (6x6) | Solve Rate (%) (6x6) |
> | :--- | :--- | :--- | :--- | :--- |
> | Gemini 2.5 Pro | 10.9 | 66.7 | 8.1 | 13.3 |
> | Grok 3 Mini | 6.7 | 40.0 | 2.9 | 6.7 |
> | DeepSeek R1 | 7.4 | 40.0 | 0.8 | 0.0 |
> | GPT 4.1 | 2.3 | 6.7 | 0.7 | 0.0 |
> | Gemini 2.0 Flash | 0.7 | 0.0 | 0.3 | 0.0 |
> | Gemma 3 27B IT | 0.1 | 0.0 | 0.0 | 0.0 |
> | Llama 4 Maverick | 0.4 | 0.0 | 0.6 | 0.0 |
> | Qwen 3 235B A22B | 8.7 | 53.3 | 1.3 | 0.0 |
> | Qwen 3 30B A3B | 4.5 | 26.7 | 0.1 | 0.0 |
> | Qwen 3 32B | 6.5 | 40.0 | 1.0 | 0.0 |
> | Qwen QwQ 32B | 5.5 | 33.3 | 0.7 | 0.0 |
>
> Due to the significant cost of certain models on multi-turn, multi-sample querying, we provide the results above for the 4x4 and 6x6 puzzles, and will prepare the 9x9 results in the final version.
>
> Moreover, we are working to add even further results to future revisions of our work, with more involved and costly strategies such as the suggested ToT.

---

### Note · Authors · 2025-08-15

We thank the reviewers for their constructive feedback and useful suggestions.

In response to the comments, we made several improvements to the paper. In summary:
- Added new evaluation results using self-consistency / majority vote in the single-step setting.
- Added additional case studies, including a detailed Gemini 2.5 Pro failure analysis, to illustrate reasoning pitfalls.
- Committed to including puzzle metadata (e.g., human difficulty ratings, solve times) to better support categorization.
- Expanded discussion with clearer definitions of creative reasoning, drawing from cognitive psychology literature.

We appreciate the reviewers’ engagement and will incorporate these updates into the camera-ready version.

---

### Decision · Program_Chairs · 2025-09-18

**Decision:**

Reject

**Comment:**

This manuscript proposed a new benchmark on long-horizon reasoning for agents based on different kinds of Sudokus. It is indeed an interesting direction to test the intelligence of the agents which also needs advanced level ability on reasoning. Most of the reviewers lean toward acceptance. However, I have a similar feeling to Reviewer zJf8. The manuscript focused too much on Sudoku design, without in-depth analysis on e.g. how to quantitatively describe the hardness of designed sudoku variants other than the grid size (as small grid size with complex rules may be harder than large grid size with no rules), how to quantitatively evaluate agent's performance other than the pass rate or number of the correct digits (as LLMs can make some random guess without really thinking about it). These information are essential for the users to leverage the data. Although the authors make some discussion period, I feel the current still need lots of polishing to be ready. Hence I lean towards rejection at this time and encourage the authors revise and re-submit to another venue.